# Interleukin-33 Involvement in Nonsmall Cell Lung Carcinomas: An Update

**DOI:** 10.3390/biom9050203

**Published:** 2019-05-25

**Authors:** Marco Casciaro, Roberta Cardia, Eleonora Di Salvo, Giovanni Tuccari, Antonio Ieni, Sebastiano Gangemi

**Affiliations:** 1School and Unit of Allergy and Clinical Immunology, Department of Clinical and Experimental Medicine, University of Messina, 98123 Messina, Italy; mcasciaro@unime.it (M.C.); gangemis@unime.it (S.G.); 2Department of Human Pathology in Adult and Developmental Age “Gaetano Barresi”, Section of Pathology, University of Messina, 98123 Messina, Italy; robertacardia87@hotmail.com (R.C.); tuccari@unime.it (G.T.); 3National Research Council of Italy (CNR), Institute of Biological Resources and Marine Biotechnologies (IRBIM), 98122 Messina, Italy; e.disalvo@isasi.cnr.it

**Keywords:** IL-33, lung, cancer, immune system, immunotherapy

## Abstract

Lung carcinogenesis is a multistep process involving genetic mutations and epigenetic changes, with the acquisition of a malignant phenotype characterized by apoptosis resistance, unregulated proliferation and differentiation, invasion, and metastatic abilities. However, neoplastic development and progression seem to be aided by non-neoplastic cells; the molecules they produced can either promote the immune response or, alternatively, support tumor pathogenesis. Consequently, the relative contribution of tumor-associated inflammatory pathways to cancer development has become crucial information. Interleukin-33 (IL-33) is an IL-1-like alarmin, and it is a ligand for the suppressor of tumorigenicity 2 (ST2) receptor. IL-33 functions as a dual role cytokine with the ability to induce T-helper-type 2 (Th2) immune cells and translocate into the nucleus, suppressing gene transcription. Although its function in immunity- and immune-related disorders is well known, its role in tumorigenesis is still debated. The IL-33/ST2 axis is emerging as a powerful modulator of the tumor microenvironment (TME) by recruiting immune cells, able to modify the TME, supporting malignant proliferation or improving antitumor immunity. In the present review, we discuss IL-33′s potential role in lung carcinogenesis and its possible application as a therapeutic target.

## 1. Introduction

Lung cancer is the first cause of cancer-associated death worldwide [1,2]. In 2015, a new World Health Organization (WHO) classification was introduced, representing the landmark for pathologists to distinguish primary lung neoplasms [3]. Therefore, for the first time, a set of standardized terms and criteria for all major histologic types of lung cancer for surgical specimens, small biopsies, and cytology was provided [4].

For diagnostic purposes, the main recommendation is to differentiate nonsmall cell lung carcinomas (NSCLC) from small cell lung carcinomas (SCLC). NSCLC have to be further classified into a more specific histologic type, such as adenocarcinoma (ADC) or squamous cell carcinoma (SCC). In detail, ADC is typically located peripherally in the lung, and some cases are linked with smoking, although it more frequently affects nonsmoking patients. Microscopically, ADC is characterized by a malignant epithelial proliferation with a glandular growth pattern and mucin production and is typically characterized by positive nuclear immunohistochemical for thyroid-transcription factor-1 (TTF-1) and membranous staining with cytokeratin-7 (CK-7) [3,4]. However, five different patterns of pulmonary adenocarcinoma have been defined, such as lepidic, acinar, papillary, micropapillary, and solid [4]. The old term brochiolo-alveolar has been abolished and replaced with the term lepidic [4]. SCC usually occurs centrally and more than 90% develops in smokers [3,4]. SCC neoplasms may be large, sometimes cavitated, and frequently appear as a polypoid mass, occluding the bronchial lumen or invading the bronchial wall. SCC has been classified into three variants: Keratinizing, nonkeratinizing, and basaloid. The use of immunohistochemistry shows a strong and diffuse immunopositivity for p63 and p40 [3,4].

In the last 2015 WHO classification, the category of neuroendocrine tumors was largely revised [3,4]. Invasive neuroendocrine tumors include three subtypes: SCLC, large cell neuroendocrine carcinoma (LCNEC), and carcinoid tumors (typical/atypical) [4,5]. SCLC typically occurs in cigarette smokers with a typical morphology characterized by cells with scant cytoplasm, round to fusiform nuclei that have a finely granular chromatin with inconspicuous or absent nucleoli, abundant necrosis, and numerous mitotic figures [3,4,5]. LCNEC is considered a high-grade tumor, usually placed in a peripheral position, with an organoid, trabecular, rosette-like architecture, and massive necrosis [4]. LCNEC elements are large, with an abundant eosinophilic cytoplasm, granular nuclear chromatin, and a high number of mitoses [3,4]. Finally, carcinoid tumors are divided into two variants: Typical and atypical. The typical variants are defined as lung carcinoid tumors measuring at least 0.5 cm, with fewer than two mitoses per 2 mm^2^ of viable area of tumor and lacking necrosis, whereas atypical ones have 2–10 mitoses per 2 mm^2^ of viable area of tumor, with the presence of necroses that are often focal [3,4,5]. All neuroendocrine tumors are defined by positive immunostaining for synaptophysin, chromogranin, and CD56 [3,4,5].

Lung carcinogenesis is a multistep process that typically takes many years to develop since several mechanisms may prevent it, such as immune and antioxidative systems, as well as DNA repair mechanisms. The recent development of new technologies has increased our knowledge of molecular carcinogenetic mechanisms, including gene amplification and protein expression, irregular cell activation, and allelic and epigenetic abnormalities [5,6]. In addition, the acquisition of multiple genetic mutations may determine a progressive development of the malignant phenotype characterized by apoptosis resistance, unregulated proliferation, angiogenesis, and metastasis [5,6].

A key role in lung carcinogenesis has also been attributed to inflammation and the action of cytokines [7,8]. Specifically, the involvement of a novel interleukin, interleukin-33 (IL-33), has emerged [7]. IL-33 is an alarmin and a member of the IL-1 family of cytokines, constitutively expressed in the nuclei of epithelial and endothelial cells, which works as a damage-associated pattern molecule to mediate tissue immune responses [2,8]. The functions of IL-33 have been widely studied in a variety of inflammatory diseases, such as asthma, rhinitis, arthritis, and inflammatory bowel disease [9,10,11]. Recent data showed the implication of the IL-33/suppressor of tumorigenicity 2 (ST2) axis in tumor development and metastasis [12]. Although it has been suggested that IL-33 is implicated in tumor-associated immune responses and inflammatory diseases of the lung [8], its role in lung cancer progression is still being debated. The aim of this review was to discuss IL-33′s possible role in lung carcinogenesis, in order to better understand its potential as a therapeutic target to reduce disease progression or to enhance the efficacy of anticancer immunotherapies.

## 2. Interleukin-33 in Lung Cancer

### 2.1. Serum Interleukin-33 in Lung Cancer Patients

Few experiments on IL-33′s involvement in tumorigenesis have been reported in the literature [13,14,15]. In detail, patients with lung cancer were reported to not show a statistical difference in the IL-33 levels and some other cytokines, like IL-27 and IL-31, compared to healthy subjects, although the patient’s cohort was small [13]. By contrast, in a larger series, serum levels of IL-33 were demonstrated to be augmented in NSCLC patients and were correlated with tumor stages [14]. These data showed that patients had reduced levels of IL-33 when compared to controls, even if they were higher in the early stages of disease [15]. 

### 2.2. Evidence of Interleukin-33 in Lung Tumor Cells 

Using immunohistochemistry, the bronchial epithelium and the vascular endothelium of normal lung tissue were shown to be reactive for IL-33 [1,2]. An evident immunopositivity was revealed in the vascular endothelium of cancerous lesions, whereas cancer cells were only stained in 30% of lung cancer patients [1,2]. The plasma IL-33 levels of these patients were not linked to the immunostaining of cancer cells [1,2]. According to these data, the bronchial epithelium and the vascular endothelium were suggested to be the source of the elevated IL-33 serum levels during the early stages of lung cancer [15]. Conversely, cell lines A549 and human pulmonary alveolar epithelial cells (HPAEpiCs) did not express the ST2 receptor and IL-33, whereas IL-1β induced IL-33 expression to the highest extent in P29 cells in vitro [16]. Activation of the IL-33/ST2 pathway was demonstrated to result in robust outgrowth and metastases of NSCLC cells [1]. Mechanistically, IL-33 signaling promotes the membrane residency of glucose transporter 1 (GLUT1), leading to increased glucose uptake and enhanced glycolysis in NSCLC cells [1]. Cancer cells are characterized by rapid and uncontrolled proliferative expansion and metastasis with high ATP demands; this shift requires a high rate of glucose uptake that greatly relies on GLUT1 activity [1]. Accordingly, in response to IL-33, NSCLC cells exert a higher glucose uptake, leading to an increased lactate production [1]. The possible mechanism may be related to activation of the nuclear factor-κB (NF-κB) by the IL-33/ST2 pathway, leading to GLUT1 membrane localization [1]. 

Recently, focus has increased on whether the role of infections in the tumorigenic effect of IL-33 determines comparable results [17]. The ability of gram-negative bacterial infections to promote NSCLC progression by a Toll-like receptor 4 (TLR4)/myeloid differentiation primary response 88 (MyD88) and IL-33 pathway may be related to increased glycolysis by the induction of GLUT1 [17]. The same infective agents could enhance lipogenesis, inducing the expression of cancer stem cell genes with a TLR4/IL-33-mediated pathway [17]. Subsequently, Wang et al. provided evidence that an IL-33 blockade efficiently limited the growth of the cancer cells [2]. IL-33 blockade using the IL-33 neutralizing antibody or the ST2 neutralizing antibody resulted in reduced proliferative survival of NSCLC cells and diminished regulatory T cells (Treg) cells in tumor tissues [2]. NSCLC-derived IL-33 supports tumor growth in an autocrine manner and educates immune surveillance in tumor microenvironments, favoring the immune escape of tumor cells [2]. An IL-33 blockade restricts NSCLC outgrowth, abrogates polarization of M2 tumor-associated macrophages (TAMs), and reduces accumulations of Treg cells in tumor tissues, thus representing an effective and promising strategy for NSCLC treatment [2]. The role of the immune system was reported once more by Saranchova et al. [18]. Firstly, they demonstrated the role of type 2 innate lymphoid cells (ILC2) in individuating and eliminating lung cancer cells, showing that neoplastic elements grew more rapidly and had a higher frequency of metastasis in IL-33 and/or ILC2-deficient mice [18]. As a result, the expression of IL-33 by the tumor may allow the activation of the ILC2 function, whereas the lack of IL-33 expression would not support these immune cells [18]. By contrast, both IL-33 and ST2 were reported to be significantly downregulated in both adenocarcinoma and squamous cell carcinoma of the lung [8]. Yang et al. successively demonstrated that IL-33 significantly promotes the migration and invasion of A549 cells by alpha serine/threonine-protein kinase (AKT) pathway activation, increasing the expression of matrix metalloproteinase-2 (MMP2) and MMP9 and facilitating the formation of metastases [12]. 

## 3. Interleukin-33 Paradox

The development of pathological angiogenesis has been implicated in both chronic inflammatory diseases and several cancers [1,2]. The main agents involved in this field are represented by growth factors, including cytokines such as interleukins [19,20,21]. Among them, IL-33 was considered a pro-cancer cytokine, since the activation pathway of IL-33/ST2 may promote metastases, as observed in some tumors, such as colorectal and ovarian cancer [19,20]. Current publications imply that vascular endothelial cells are the dominant IL-33-expressing cell population in vivo, promoting angiogenesis and endothelial permeability in an endothelium-derived nitric oxide (NO)-dependent manner [22]. However, the role of the IL-33/ST2 axis has recently been related to cancers, in which the deletion of ST2 signaling may enhance the antitumor immune response [23]. IL-33 is now classified as a member of the IL-1 cytokine family that induces T-helper-type 2 (Th2)-associated cytokines [24]. IL-33 is released from damaged cells and therefore likely plays a role similar to that of IL-1 and high-mobility group box 1 (HMGB1). These latter cytokines are called alarmins because they activate the immune system in response to trauma or infections [7,25]. IL-33 certainly enhances the development of Th2-associated diseases [7], but it has been revealed to induce T helper type 1 (Th1)/Th17 immunity [26,27]. Depending on the environmental conditions, IL-33 may orchestrate antitumor immunity, activating CD8-positive T cells [26,27]. The Th1/Th2 paradigm has been suggested to act in tissues surrounding lung cancers, since Th2 cell development is triggered by tumor antigens and IL-12/ interferon-γ (INF-γ) [28]. These latter molecules activate the tumor-specific CD8+ CTL cells required for the elimination of cancer cells [29]. Th17 and Treg cells are becoming recognized as novel immune modulators in lung cancer [30]; in particular, Th17 cells modulate antitumor immune responses directly or via the activated production of proinflammatory cytokines [30]. Treg cells may play important roles in the preservation of self-tolerance and in the modulation of the overall immune response to tumor cells [31]. Wang et al. demonstrated that Treg cells are influenced by IL-33 [2]. They determined the reduction of Treg cells in experimental lung cancer after administering anti-IL-33 and anti-ST2 [2]. An additional effect may be produced by IL-33 on ILC2 [32]. Specifically, using an autocrine mechanism, this alarmin secreted by the primary tumor seems to stimulate ILC2 to recruit Th1 cells, which in turn enroll cytotoxic cells in an attempt to kill malignant cells [32]. IL-33 could inhibit tumor progression by recruiting CD8+ T, natural killers (NKs), eosinophils, and dendritic cells, and by the induction of IFN-γ [32]. Therefore, a dual paradoxical role should be attributed to IL-33 (Figure 1). 

## 4. Interleukin-33 Multiple Pathways

Among the hypotheses to clarify the ability of IL-33 to start or sustain the oncogenic process, the stimulation of the extracellular signal-regulated kinase (ERK) and the c-Jun N-terminal protein kinase (JNK) pathways may be determined by the abovementioned alarmin [20,33,34]. The cancer Osaka thyroid oncogene (COT) is another kinase activated by IL-33, leading to tumor progression [35]. However, the centrality of IL-33 in the metastatic process was highlighted by the experiment of Sun et al. [17]. They observed that gram-negative bacterial infections in patients affected by NSCLC predicted a worse prognosis [17]. In detail, this infection over-expressed IL-33, involving the MyD88 pathway by activating TLR4 signaling [17]. HMGB1, another alarmin, binds to the lipopolysaccharide (LPS) of gram-negative bacteria. Through the link between LPS and TLR4, a series of pro-inflammatory cytokines, NF-κB included, alert the innate immune system. Finally, gram-negative bacteria, by augmenting the IL-33, which in turn leads to GLUT1 over-expression and the production of lactate, favor malignant cell overgrowth [17]. IL-1β was also reported to be able to increase IL-33 P29 cells. Some studies reported that the alarmin, after the binding to ST2, activated the AKT pathway, leading to immune cells recruitment, cancer development, and the stimulation of metalloproteinases, which are known to facilitate metastasis diffusion [16]. The IL-33/ST2 axis is also responsible for the NF-κB release, which augments GLUT1 expression in NSCLC. The administering of anti-IL-33 and anti-ST2 is effective in limiting this process and in diminishing the presence of Treg cells at the site of cancer [2]. 

Respiratory epithelial cells constitute a main source of both CC and CXC chemokines and are fundamental gears in regulating the immune and inflammatory “lung machine”. In this scenario, tumor necrosis factor tumor necrosis factor (TNF)-α and IL-1β are some of the most powerful inducers of chemokines in the lung environment [36]. IL-18 and IL-33, which belong to the IL-1 family, still have a controversial role in the activation of epithelial cells. Although they belong to the same family with a common molecular composition which leads to similar biological functions, they still have some differences as a consequence of their stimulation [36]. Both promote the recruitment of Th2-associated cytokines, being proper Th2 chemo-attractants [36]. Chemokines released in inflammatory conditions from lung epithelial cells could act as key players for the recruitment of neutrophils, monocytes, NK cells, dendritic cells, mast cells, or T lymphocytes directly at the site of damage. For this reason, new pharmacological approaches should involve chemokine antagonists to interrupt the connection between epithelial cells and immune cells [36].

## 5. Concluding Remarks 

In our opinion, there is no doubt that IL-33 has a dual role in lung cancer. In most cases, IL-33 is overexpressed in tumor cells, whereas, conversely, a high quantity rate of alarmin in patients’ blood coincides with reduced tumor progression. IL-33, together with its receptor ST2, appears to have a central role in the tumorigenesis process by activating several pathways, such as interferon regulatory factor-3 (IRF-3), MyD88, AKT, COT, ERK, and JNK. IL-33 may act through an increase in cells’ metabolism and a regulation of the immune system, in both a pro- and anti-oncogenic way. Further studies should focus on these activation patterns and on the effects induced by the administration of anti-IL33/ST2 antibodies to establish a basis for future therapeutic options.

## Figures and Tables

**Figure 1 biomolecules-09-00203-f001:**
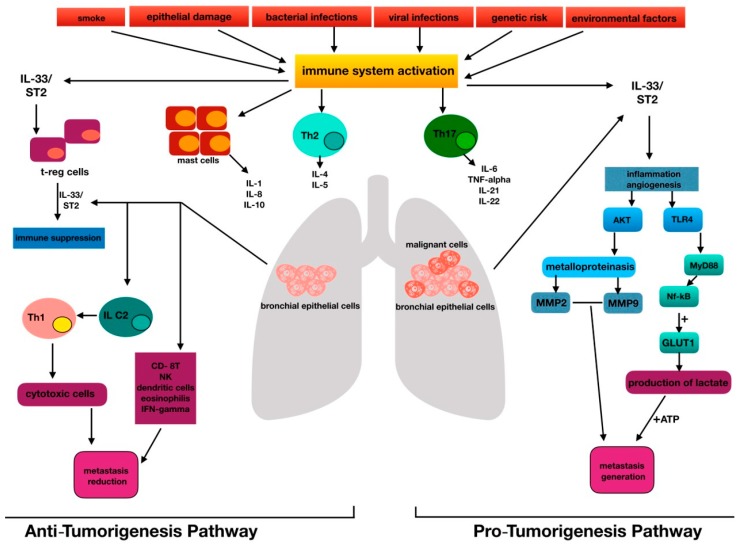
The figure represents the interleukin-33 (IL-33) paradox, highlighting the centrality of IL-33 in the pathogenesis of lung cancer for immune system regulation, metabolic activity, and the metastasis genesis process.

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
