# Peer review of "Interleukin-33 Involvement in Nonsmall Cell Lung Carcinomas: An Update"

_biomolecules, 2019, doi:10.3390/biom9050203_

Round 1
Reviewer 1 Report
The authors present a short review on the research that has been conducted examining the role of the IL33/ST2 pathway in the pathogenesis of lung cancer. Although the number of studies in this area is not extensive, they provide evidence on the contrasting roles of IL-33 in lung neoplasms. In some cases, they proposed tumor-promoting effects of IL-33 while in others, an anti-tumorigenic effect.
The outstanding main issue that still needs to be addressed is that there are numerous grammatical errors throughout the manuscript that makes it confusing to read in many places. These should have been corrected in earlier drafts, but they were not. I have no confidence that the authors respect my comments. My recommendation is to reject. Given the conciseness of this review it is of high importance to have the overall ideas conveyed in the best manner possible.
Some of the outstanding corrections that remain unchanged:
There is no reference of the figures in the text.
Line 27: Change “Actually” for Interestingly
Line 29-30: Change sentence to “The aim of this review is to discuss the possible role of IL-33 in lung…”
Line 40-54: Need to re-write section to better present the idea. The sentences need to better connect.
Line 55-68: Need to use more references to support statements since only using ref #4. As this is a review there should be multiple sources to support the presented idea or information.
Line 55: What year was the last WHO classification?
Line 77-78: Need reference for statement “A key role…action of cytokines.”
Line 86: Change to ‘’…cancer progression is still not well elucidated.”
Line 91: Need to change wording “As it emerged from literature…”
Line 92-93: Change to “…patients with lung cancer do not show statistical difference in the IL-33 levels…”
Line 94,96: Change to Hu et al. and Kim et al. This also need to be done for other sections in this review in which the author is stated in the text in
Line 95: Change to “…that serum levels of IL-33 were…”
Line 96-97: Change to: “… showed that patients had reduced levels of IL-33 when compared to controls, even if…”
Line 116-117: Change to “Recently there has been an increased focus on the role…”
Line 133: Change to “In contrast, it has been reported that both…”
Line 139-140: Change to “… has been implicated in both chronic…”
Line 142-143: Change to “…IL-33/ST2 may promote metastases…”
Line 143: Change to “…colorectal and ovarian cancer…”
Line 159: Change to “…Th17 cells both modulate antitumor…”
Line 162: Change to “Wang et al. demonstrated that Treg were influenced by IL-33.”
Line 175: Figure 1b should be named to figure 2 and a figure legend should be added
Line 178: Change to “…has been proposed that the stimulation…”
Line 188: Change to “…production of lactate, favor malignant cell overgrowth.”
Line 194-195: Need to change wording “As much as emerge from literature…”
Author Response
In relation to Your suggestions we have performed all requested corrections, regarding lines 27, 29,30,40,54, 55-68, 77-78, 86,91, 92-93, 94,96, 95-97,116-117, 133, 139-140, 142-143, 159, 162, 172, 175, 178, 188, 194-195. All changes were highlighted by red ink.
Finally an extensive professional English editing has been done by MDPI scientific translation service.
Reviewer 2 Report
While there has been some improvement in the manuscript from the first submission, it is still poorly written with many instances where the statements are vague and very confusing [lines 91-92, 99-105 (this whole paragraph could be rewritten to make it better), 116-117 is confusing as well, among many others]. Some sentences are overstated (paragraph 1 of Introduction, lines 34-36). Again, this reviewer strongly suggests that the manuscript be reviewed and edited by someone with a good command of the English language. At it's current form, I do not recommend publication.
Author Response
In relation to Your suggestions we have performed all requested corrections, regarding lines 91-92, 99-105, 116-117, paragraph 1 of Introduction, lines 34-36). Finally an extensive professional English editing has been done by MDPI scientific translation service.
Round 2
Reviewer 1 Report
Paper is acceptable in present form.